# Putnam-AXIOM: A Functional & Static Benchmark for Measuring Higher Level Mathematical Reasoning in LLMs

Aryan Gulati [* 1]   Brando Miranda [* 1]   Eric Chen [* 1]   Emily Xia [* 1]   Kai Fronsdal [* 1]   Bruno Dumont [1]
Sanmi Koyejo [1]

## Abstract

Current mathematical reasoning benchmarks for large language models (LLMs) are approaching saturation, with some achieving $> 90\%$ accuracy, and are increasingly compromised by training-set contamination. We introduce Putnam-AXIOM, a benchmark of 522 university-level competition problems drawn from the prestigious William Lowell Putnam Mathematical Competition, and Putnam-AXIOM Variation, an unseen companion set of 100 functional variants generated by programmatically perturbing variables, and constants. The variation protocol produces an unlimited stream of equally difficult, unseen instances – yielding a contamination-resilient test bed. On the Original set, OpenAI's o1-preview – the strongest evaluated model – scores 41.9%, but its accuracy drops by 19.6 % (46.8% relative decrease) on the paired Variations. The remaining eighteen models show the same downward trend, ten of them with non-overlapping 95% confidence intervals. These gaps suggest memorization and highlight the necessity of dynamic benchmarks. We complement ("boxed") accuracy with Teacher-Forced Accuracy (TFA), a lightweight metric that directly scores reasoning traces and automates natural language proof evaluations. Putnam-AXIOM therefore provides a rigorous, contamination-resilient evaluation framework for assessing advanced mathematical reasoning of LLMs. Data and evaluation code are publicly available at https://github.com/brando90/putnam-axiom.

*Equal contribution   [1]Department of Computer Science, Stanford University, Stanford, USA. Correspondence to: Aryan Gulati <aryangul@cs.stanford.edu>, Brando Miranda <brando9@cs.stanford.edu>, Sanmi Koyejo <sanmi@cs.stanford.edu>.

*Proceedings of the 42$^{nd}$ International Conference on Machine Learning*, Vancouver, Canada. PMLR 267, 2025. Copyright 2025 by the author(s).

## 1. Introduction

The ability for Large Language Models (LLMs) to reason about complex problems has a plethora of applications in fields such as economics (Zhang et al., 2024), drug discovery (Bran et al., 2023), and even simulations of human behavior and society (Park et al., 2023). Rapid adoption of LLMs for reasoning has, in turn, spurred rapid gains on standard reasoning benchmarks (OpenAI, 2023; Gemini, 2023; Yang et al., 2024a).

**Outpacing Current Evaluations.** Indeed, advanced models like GPT-4 (OpenAI, 2023) and Gemini Ultra (Gemini, 2023) have reported human-level performance on many benchmarks like MMLU (Hendrycks et al., 2020) and MMMU (Yue et al., 2023). Similarly, LLMs have seen progress in other challenging benchmarks like GSM8K (Chen et al., 2022) and MATH (Hendrycks et al., 2021), with SOTA models attaining nearly 90% accuracy on MATH (Lei, 2024) and nearly perfect accuracy on GSM8K (Zhong et al., 2024). Although this progress demonstrates rapidly evolving LLM capabilities, it creates an evaluation ceiling effect: current benchmarks lack sufficient difficulty to discriminate between state-of-the-art models' reasoning abilities.

**Data Contamination.** The problem is further complicated by data contamination, which remains a major concern for current evaluation benchmarks. By training LLMs on larger portions of the internet, researchers are incorporating an increasing number of open-source benchmark data into the models' pretraining. Therefore, a model can display artificially high "reasoning ability" by simply memorizing the answers it has seen, undermining the integrity of the evaluation.

To address these limitations, we introduce the Putnam-AXIOM (**A**dvanced e**X**amination of **I**ntelligence in **O**perational **M**athematics) dataset, a novel and challenging compilation of high-level mathematics problems sourced from the William Lowell Putnam Mathematical Competition, an annual mathematics competition for undergraduate college students in North America which requires advanced mathematical reasoning and covers a wide range of university-level mathematical concepts. In addition, we

also introduce functional variations of the Putnam-AXIOM dataset to combat data contamination, taking inspiration from the solution employed by (Srivastava et al., 2024). Functional variations adjust variables, constants, and the phrasing of problems through Python scripts, allowing us to generate an unlimited number of new problems that are not found on the Web but still retain their mathematical complexity and validity. Putnam-AXIOM enables fully automated evaluations by requiring models to provide final answers within "\boxed{}" brackets which can then be extracted and compared to the ground truth final solution using an equivalence function [1] as used for the MATH dataset (Hendrycks et al., 2021). This approach eliminates the need for human evaluation, and avoids the limitations of multiple-choice formats (Schaeffer et al., 2024), thus maintaining soundness while enabling scalability.

Initial evaluations on Putnam-AXIOM demonstrate its difficulty with o1-preview scoring less than half at 41.94%, while GPT-4o achieves only 19.35%. Even math-specialized models such as Qwen2-Math-7B and Qwen2-Math-7B-Instruct perform poorly, scoring 5.51% and 11.8% respectively. Performance further declines on functional variations of Putnam-AXIOM, which include significant drops for most models, decreasing by 20-30% in relative performance. "These low accuracies demonstrate Putnam-AXIOM's effectiveness as a challenging benchmark, while variations expose models' reliance on memorization.

**Proof-based Evaluation Metrics.** In addition to introducing the Putnam-AXIOM Original and Variation benchmarks, we identified the need for more sophisticated LLM reasoning evaluation metrics. Current evaluation metrics for reasoning are inadequate, as they rely solely on a final "boxable" answer without assessing the actual reasoning process. For problems with only a few possible final answers – such as with true/false or modular arithmetic – this means models will often get the final boxed answer correct by random chance. Additionally, evaluations are then limited to the subset of problems with simple boxable answers, completely ignoring theorem proving and problems with complex formulae as the solution. In open-ended evaluations areas, like theorem proving, the current standard is either human evaluations (He et al., 2024), which are expensive, or using a formal language, which requires translating theorems and setting up complex environments and dependencies (Yang et al., 2024b). We therefore explore alternative automatic metrics to boxed answers and find that a simple and cheap method, Teacher-Forced Accuracy (TFA), is a promising approach.

Our **contributions** are:

- The **Putnam-AXIOM**, a new evaluation benchmark of 522 challenging mathematical problems sourced from the William Lowell Putnam Competition, designed to assess advanced mathematical reasoning in LLMs.
- **Functional variations** for 100 of these problems using Python scripts, altering variables, constants, and problem phrasing to generate unlimited novel problems while preserving their mathematical complexity, effectively avoiding data contamination.
- Teacher-forced accuracy (TFA), to provide a more **complete assessment** of LLMs' reasoning abilities – *beyond* traditional boxed answers.

## 2. Related Work

### 2.1. Mathematics benchmarks

Numerous benchmarks exist to assess the mathematical capabilities of models, each typically focusing on a specific task. Two notable examples are MATH (Hendrycks et al., 2021) and GSM8K (Cobbe et al., 2021). The MATH dataset contains questions sourced from American high school mathematics competitions such as the AMC 10, AMC 12, and AIME (Hendrycks et al., 2021), while the GSM8K dataset contains 8.5K handwritten elementary school level questions (Cobbe et al., 2021). Both contain questions and answers with detailed rationale explanations.

As models have become larger and more powerful, even the most difficult existing benchmarks have become less challenging. For instance, while the MATH dataset saw $6.9\%$ accuracy on its release, it now sees $87.92\%$ accuracy with GPT-4 MACM (Lei, 2024). Similarly, GPT4 has attained $97.1\%$ accuracy on the GSM8K (Zhong et al., 2024). This saturation necessitates the development of more challenging benchmarks.

Many contemporary data sets have been created to combat the saturation of existing benchmarks. For instance, the ARB dataset includes hundreds of challenging problems in high school and college-level math, physics, and chemistry (Sawada et al., 2023). Similarly OlympiadBench contains nearly 9,000 problems from the International Mathematics Olympiad (IMO), the Chinese GaoKao, and more (He et al., 2024). Finally, SciBench is a similar reasoning benchmark that includes hundreds of college-level scientific reasoning questions from instructional textbooks (Wang et al., 2023).

Although these datasets alleviate the saturation problem, they come with many limitations. For instance, ARB (Sawada et al., 2023) and OlympiadBench (He et al., 2024) both contain several symbolic and proof-based questions which cannot be graded automatically and require a costly and lengthy human evaluation process. Though ARB attempts to utilize LLMs to grade their own responses with a rubric, this process is often unreliable and self-referential

---

[1]For instance, the equivalence function would evaluate the answers 0.5, 1/2, and \frac{1}{2} as equal.

(Huang et al., 2024a). Our Putnam-AXIOM dataset addresses these limitations by offering challenging Putnam problems with fully-written solutions and easily evaluable answers. It enables efficient automated assessment via frameworks like LM Harness (Gao et al., 2024), avoiding costly human evaluation or unreliable self-grading.

PutnamBench (Tsoukalas et al., 2024) is a related benchmark that primarily focuses on formal theorem proving. Its main objective is to derive formalized proofs of mathematical statements and it provides formalizations in systems such as Lean, Isabelle, and Coq, all sourced from the prestigious Putnam competition. PutnamBench also includes 640 natural language statements and their corresponding answers where applicable. While both benchmarks draw from the same competition, Putnam-AXIOM focuses on the curation of natural language problems for final answer verification and introduces automatic functional variations to generate additional benchmarks addressing potential data contamination. Further through Putnam-AXIOM we go beyond just the final answer by assessing the model outputted solution through evaluation proxy metrics.

## 2.2. Functional Benchmarks

Data contamination is a significant problem in creating evaluation benchmarks, as many of these problems are openly available on the Internet and are likely included in the training data for large models (Schaeffer, 2023; Sainz et al., 2023). Thus, the MATH (Hendrycks et al., 2021), AGIEval (Zhong et al., 2023), OlympiadBench (He et al., 2024), and ARB (Sawada et al., 2023) benchmarks (which are all sourced from problems on the Internet) could potentially be contaminated. Therefore, models may achieve artificially high performance on an evaluation benchmark by memorizing the answers to the problems Magar & Schwartz (2022); Ranaldi et al. (2023).

A straightforward way of avoiding data contamination issues is to utilize problems unavailable on the Internet. However, even if problems are not currently part of model training data, it is unrealistic to expect them to remain inaccessible. At the same time, it is costly to rely on the continuous human development of new datasets.

Srivastava et al. (2024) attempts to alleviate this data contamination issue by creating *functional* variations of the MATH dataset, where new problems can be generated simply by changing numeric parameters, yielding different solutions. They observe a significant discrepancy in models' performance between standard benchmarks and these new variations. We recognize the potential of this idea and have adapted it to our more challenging dataset. We have altered the variables, constants, and phrasing of many Putnam questions while preserving their overall difficulty and requirements for logical and mathematical reasoning.

## 2.3. Evaluation Metrics

Several approaches have been proposed to reduce the reliance of model evaluations on box-able answers, particularly in domains like free-form writing or translation where unique answers do not exist (Leiter et al., 2022; Opitz & Frank, 2021). Historically, tasks such as translation and natural language generation, which lack a single correct answer, have used more flexible metrics, including *n*-gram match (Lin, 2004), model-based (Guerreiro et al., 2023), embedding proximity (Zhang et al., 2020), paraphrasing (Thompson & Post, 2020), generation as an evaluator (Yuan et al., 2021), and information alignment (Deng et al., 2021). However, these metrics are not designed to assess reasoning ability or the correctness of mathematical statements.

When relying on boxed answers, we simply do not know how often the generated reasoning steps actually support the final answer. For evaluating reasoning abilities, the ROSCOE suite of metrics is noteworthy as it measures various fine-grained aspects of reasoning steps such as semantic consistency, logicality, informativeness, fluency, and factuality (Golovneva et al., 2023). We omit descriptions of each metric, but highlight that most of them rely on sentence embedding models and operate on a step-by-step level. Unfortunately, the original ROSCOE metrics were predominantly tested on GPT-3 generations, and we find that these metrics do not provide evaluations that are comparable across different models. Although fine-grained metrics like ROSCOE can be useful for interpreting specific aspects of a model's capabilities, an ideal reasoning benchmark would employ a single metric that is comparable across models and highly correlated with the correctness of the generated reasoning.

In Huang et al. (2024b), authors drew upon equivalence between language modelling and compression. They demonstrated that using bits per character (BPC) to measure a model's compression rate on several external large corpora is highly correlated with model performance on various benchmarks. However, this approach has drawbacks: evaluating compression on large corpora is expensive, and the equivalence only holds for base models, as fine-tuned models are not general-purpose compressors for arbitrary text. Despite this, we suspect there would still be a relatively high correlation for most fine-tuned models. Relatedly, Yuan et al. (2023) found that pre-training loss is strongly correlated with mathematical ability for the LLaMA family (Touvron et al., 2023a;b). Unfortunately, creating an open benchmark using this metric is impractical due to the dependence of pre-training loss on differences in pre-training data, tokenizers, and other training-specific parameters.

Relationship to process-supervision metrics. Teacher-Forced Accuracy (TFA) complements recent step-level evaluation methods based on process supervision. PRM-style approaches label each intermediate step with a learned re-

ward model trained on either expert annotations (Lightman et al., 2023) or large-scale automated traces (Luo et al., 2024a). While PRMs are powerful, they require (i) millions of step-level labels, (ii) an additional model to learn the reward, and (iii) non-trivial calibration at inference time. By contrast, TFA dispenses with reward learning entirely: given a reference proof, we condition the LLM on the gold prefix and measure whether it predicts the next step. This teacher-forcing procedure yields a direct, noise-free estimate of reasoning fidelity, is agnostic to model size, and incurs only a single forward pass per step. Empirically, we find that TFA correlates with final-answer accuracy but, like PRM scores, can still penalize solutions that "get the right box" via spurious reasoning, thereby providing a lightweight yet alternative for ("boxed") final answer accuracy.

## 3. Methods

### 3.1. Putnam-AXIOM Original Dataset

**Dataset.** The Putnam-AXIOM Original Dataset contains 522 problems curated from the William Lowell Putnam Mathematical Competition posed between 1938 and 2023. These problems were selected based on their ability to yield a unique, numerically evaluable final answer, enabling automated assessment while preserving mathematical rigor. The dataset encompasses various topics within university-level mathematics categorized into 11 distinct domains – Geometry, Algebra, Trigonometry, Calculus, Linear Algebra, Combinatorics, Probability, Number Theory, Complex Numbers, Differential Equations and Analysis.

To maintain a consistent and rigorous evaluation, each problem retains its original exam ID, which indicates its difficulty level and the topic categories. The ID format includes the exam sitting (A or B) and a number (1-6) representing increasing complexity, with 1 being easiest and 6 being most difficult. The dataset is formatted using LATEX to accurately capture the complex equations and symbols the problems employ. Additionally, we utilize Asymptote vector graphics for encoding mathematical figures and diagrams to ensure language models can process visual elements directly. Further, we standardized the placement of boxed answers by relocating them to the end of each solution string to minimize unintended emergent behaviors leading to evaluations that are less "harsh" or prone to penalizing the model for formatting deviations rather than actual comprehension.

**Modified Boxing.** Given the complex nature of certain Putnam questions, some problems do not lend themselves to simple, singular boxed final answers. Instead, they often include conditions, multiple possible answers, varied answer formats and elaborate proofs. These original questions would have necessitated costly and difficult human evaluations which we seek to avoid. To address this, we modified these questions by adding a trivial next step to the original questions, changing the solution accordingly. This additional step was designed so as to ensure that solvers reached the same conclusions and insights necessary to solve the problem, but then output a single boxed final answer. We provide an example of such a change in Figure 1. By incorporating this minor modification, we preserved the inherent difficulty and complexity of the original problems while making the answers suitable for automated evaluation. Furthermore, since Putnam proof-based problems often test different reasoning abilities than Putnam answer-based problems, modified boxing allows us to provide a more comprehensive test. Of the original 522 problems, 221 required modified boxing, representing 42.3% of the dataset.

### 3.2. Putnam-AXIOM Variation Dataset

Models trained on snapshots of the internet have likely encountered Putnam questions, potentially inflating their performance on the Putnam-AXIOM Original dataset. Therefore, drawing inspiration from (Srivastava et al., 2024), we introduce functional variations of select problems from Putnam-AXIOM Original providing an effective way of evaluating models that have been trained on the entire internet by taking advantage of weaknesses in model memorization. These variations are classified into two types.

**Variable Change.** The simplest variation is a variable change, where variable names are altered and the final answer is unvaried. Variable changes slightly modify the problem from its original statement, which models could have trained on.

**Constant Change.** Constant changes modify numeric properties of the question, altering constants within the step-by-step solution and the final answer. Constant changes significantly transform the problem from its original statement, challenging models to perform complex reasoning on how the changes affect the solution and final answer, as in the example from Figure 2.

**Variational Dataset Description.** We created functional variations for 100 Putnam-AXIOM questions (19.2% of the full dataset), selected to maximize coverage across mathematical domains while ensuring variation feasibility (problem-specific constants, non-generalizable solutions, and questions lacking constants or boxable answers were left). The dataset includes 37 constant+variable and 63 variable-only changes. We rephrased problem statements while maintaining the core task to prevent pattern recognition by LLMs. Each variation can generate infinite unique, equally difficult snapshots, offering a sustainable evaluation method. To evaluate various SOTA models, evaluators are expected to generate snapshots (instances of the infinite potential variations) of the variation dataset by running the

**Problem:** Determine which positive integers $n$ have the following property: For all integers $m$ that are relatively prime to $n$, there exists a permutation $\pi \colon \{1, 2, \ldots, n\} \to \{1, 2, \ldots, n\}$ such that $\pi(\pi(k)) \equiv mk \pmod{n}$ for all $k \in \{1, 2, \ldots, n\}$.

**Solution:** The desired property holds if and only if $\boxed{n = 1 \text{ or } n \equiv 2 \pmod 4}$. Let $\sigma_{n,m}$ be the permutation of $\mathbb{Z}/n\mathbb{Z}$ induced by multiplication by $m$; the original problem asks for which $n$ does $\sigma_{n,m}$ always have a square root.

$\ldots$

By Lemma 1, $\sigma_{n,m}$ does not have a square root.

**Year:** 2016     **ID:** A1     **Final Answer:** ??

**Problem:** Determine the sum of the first $k$ positive integers $n$ (in terms of $k$) which have the following property: For all integers $m$ that are relatively prime to $n$, there exists a permutation $\pi \colon \{1, 2, \ldots, n\} \to \{1, 2, \ldots, n\}$ such that $\pi(\pi(k)) \equiv mk \pmod{n}$ for all $k \in \{1, 2, \ldots, n\}$.

**Solution:** Let $\sigma_{n,m}$ be the permutation of $\mathbb{Z}/n\mathbb{Z}$ induced by multiplication by $m$; the original problem asks for which $n$ does $\sigma_{n,m}$ always have a square root.

$\ldots$

The desired property holds if and only if $n = 1$ or $n \equiv 2 \pmod 4$, hence making the required sum $\boxed{2k^2 - 4k + 3}$.

**Year:** 2016     **ID:** A1     **Final Answer:** $2k^2 - 4k + 3$

*Figure 1.* **A modified boxing example in Putnam-AXIOM**. Here we see that the original problem holds true for a number of values of $n$ conditioned on a specific property making it hard to find a boxable expression. We thus modify the solution to still require the solver to get to that conclusion and add a further computation of summing up the first $k$ such values of $n$ giving a boxable solution while keeping the core of the problem the same.

generation code.

### 3.3. Model Evaluations

Using the LM Harness Evaluation framework (Gao et al., 2024), we evaluated several open-source and proprietary SOTA LLMs. We rely on LM Harness Evaluation because its widely-used, vetted codebase that lets us reuse the same tokenizer-aware boxing extractor and MATH (Hendrycks et al., 2021) equivalence function, giving our results an immediately reproducible and trustworthy evaluation pipeline. Models were prompted to provide answers in \boxed format, which were then compared to Putnam ground truths with an exact final answer match. We evaluated the 522-question Putnam-AXIOM Original dataset once. For the variation dataset, we conducted five trials, each using a randomly selected variation snapshot and its corresponding 100 original questions. We then calculated mean accuracy and 95% confidence intervals.

### 3.4. Fine-Tuning Experiments with Variations

To simulate data contamination and evaluate the robustness of the Putnam-AXIOM Variations dataset, we performed LoRA fine-tuning using the next-token prediction objective and trained until convergence. The models were then evaluated before and after fine-tuning on a 100-question subset of the Putnam-AXIOM Variations benchmark and its corresponding original problems. Prior to fine-tuning, the model achieved 12% accuracy on the variation set and 23% on the originals. After fine-tuning, accuracy on the original ques-

tions rose sharply to 80%, while accuracy on the variations increased only modestly to 33%. These results suggest that the model rapidly memorized the original problems while continuing to struggle with functionally equivalent variations, highlighting the importance of contamination-resilient benchmarks.

### 3.5. Proxy Reasoning Metrics

Final-answer ("boxed") accuracy treats an LLM's chain of thought as a black box: it cannot penalize lucky guesses on binary questions, detect spurious derivations, or compare the quality of two correct proofs. Recent *process-supervision* work mitigates this by training reward models that score every intermediate step, but requires vast step-level annotations and an extra model at inference time (**?**Luo et al., 2024b). To obtain step-level insight *without* additional annotation, we introduce **Teacher-Forced Accuracy (TFA)** — a deterministic metric that measures how well a model predicts each reference step under teacher forcing. We then compare TFA and its relatives with the 18 automatic metrics in ROSCOE (Golovneva et al., 2023).

**Teacher Forcing:** In teacher forcing (Jiang et al., 2023; Lamb et al., 2016), the model is conditioned on the ground truth solution tokens rather than its own previous predictions. Given a question $q$ and its ground truth solution tokenized as $s_1, s_2, \ldots, s_N$, let $\hat{s}_1, \hat{s}_2, \ldots, \hat{s}_N$ be the tokens predicted by the model under teacher forcing. We explore the following teacher forcing metrics:

**Problem:** Define a *growing spiral* in the plane to be a sequence of points with integer coordinates $P_0 = (0,0), P_1, \ldots, P_n$ such that $n \geq 2$ and:

$$\ldots$$

How many of the points $(x, y)$ with integer coordinates $0 \leq x \leq 2011, 0 \leq y \leq 2011$ *cannot* be the last point, $P_n$ of any growing spiral?

**Problem:** Consider a *growing spiral* in the plane, defined as a sequence of points $L_0 = (0,0), L_1, \ldots, L_n$, each having integer coordinates, where $n \geq 2$ and:

$$\ldots$$

Determine the number of points $(w, v)$ with integer coordinates $0 \leq w \leq 4680, 0 \leq v \leq 4680$ that *cannot* be the final point, $L_n$ of any such growing spiral.

**Solution:** We claim that the set of points with $0 \leq x \leq 2011$ and $0 \leq y \leq 2011$ that cannot be the last point of a growing spiral are as follows: $(0, y)$ for $0 \leq y \leq 2011$; $(x, 0)$ and $(x, 1)$ for $1 \leq x \leq 2011$; $(x, 2)$ for $2 \leq x \leq 2011$; and $(x, 3)$ for $3 \leq x \leq 2011$.

$$\ldots$$

This gives a total of

$$2012 + 2011 + 2011$$

$$+2010 + 2009 = \boxed{10053}$$

excluded points.

**Solution:** We claim that the set of points with $0 \leq w \leq 4680$ and $0 \leq v \leq 4680$ that cannot be the last point of a growing spiral are as follows: $(0, v)$ for $0 \leq v \leq 4680$; $(w, 0)$ and $(w, 1)$ for $1 \leq w \leq 4680$; $(w, 2)$ for $2 \leq w \leq 4680$; and $(w, 3)$ for $3 \leq w \leq 4680$.

$$\ldots$$

This gives a total of

$$4681 + 4680 + 4680$$

$$+4679 + 4678 = \boxed{23398}$$

excluded points.

**Year:** 2011  **ID:** A1  **Final Answer:** 10053

**Year:** 2011  **ID:** A1  **Final Answer:** 23398

*Figure 2.* **Constant and variable change in Putnam-AXIOM.** Here, we perform a variable change on the original problem/solution on the left by changing variables '$x$' to '$w$,' '$y$' to '$v$,' and '$P$' to '$L$.' We also perform a constant change by altering the constant '2011' to '4680'. The constant change affects the final answer, changing it from 10053 to 23398. Finally, we rephrase the problem.

1. Teacher-Forced Accuracy (TFA) measures the proportion of tokens that the model predicts correctly when conditioned on the ground truth tokens.

$$\text{TFA} = \frac{1}{N} \sum_{i=1}^{N} \mathbb{1}[\hat{s}_i = s_i]$$

2. Teacher-Forced Cross Entropy (TFCE) measures the average negative log likelihood of the ground truth tokens under the model's predicted probability distribution.

$$\text{TFCE} = -\frac{1}{N} \sum_{i=1}^{N} \log \mathbb{P}(\hat{s}_i = s_i \mid q, s_1, s_2, \ldots, s_{i-1})$$

3. Perplexity is a measure of how well a probability distribution predicts a sample. In the context of teacher forcing, it is an exponentiation of the cross entropy.

$$\text{Perplexity} = \exp(\text{TFCE})$$

4. Bits Per Character (BPC) (Huang et al., 2024b) is very similar to TFCE and has been shown to correlate well with benchmarks when evaluated on very large corpora. The idea is that due to differences in tokenization, average bits per token are not directly comparable. Instead we use

$$\text{BPC} = -\frac{1}{T} \sum_{i=1}^{N} \log \mathbb{P}(\hat{s}_i = s_i \mid q, s_1, s_2, \ldots, s_{i-1})$$

where $T$ is the number of characters in the solution string rather than the number of tokens.

The main limitation of the teacher forcing approach is the dependency on the ground truth solution. Models are often finetuned for a specific style or problem solving approach (such as tool use or code generation). In this case, we would expect that teacher forcing metrics would under represent the models' abilities.

**ROSCOE**: The ROSCOE suite offers 18 distinct metrics, each tailored to assess a different facet of reasoning as described by (Golovneva et al., 2023). These metrics are broadly categorized into four groups. The first category, semantic alignment, focuses on identifying relationships between concepts that share the same or similar meanings. Metrics in this category typically examine reasoning on

a step-by-step basis. In contrast, semantic similarity metrics evaluate the problem and solution holistically. Logical inference metrics, utilizing a specially trained model (Laurer et al., 2024), detect contradictions between reasoning steps. Lastly, language coherence is assessed by evaluating model outputs using the perplexity score from GPT-2 Large (Radford et al., 2019) and a grammar model (Krishna et al., 2020). We use the code provided by the authors as is to evaluate these metrics.

**Metric Evaluation:** Given the challenging nature of Putnam-AXIOM and the poor performance of existing models, we opted to test the proposed proxy metrics on the MATH dataset instead. For a metric to be effective as a benchmark, its evaluations must be comparable across different models. To generate evaluation data, we utilized 15 open-source models, ranging from 7 billion to 70 billion parameters, which exhibit a wide range of performance across the 7 different MATH datasets. We then compared the proxy metric evaluations with each model's boxed accuracy for each dataset. A high correlation between the proxy metric and boxed accuracy indicates a better proxy.[2] Our results, including the raw correlations for each metric in Table 4, are presented in the Appendix.

## 4. Results

### 4.1. Putnam-AXIOM Model Performance

Table 1 presents Putnam-AXIOM Original dataset accuracies. Most models score below 10%, with even NuminaMath, the AI Mathematics Olympiad winner (Investments, 2024), achieving only 10.34%. These low accuracies demonstrate Putnam-AXIOM's effectiveness as a challenging benchmark. Figure 3 contrasts Putnam-AXIOM Variation dataset mean accuracies with the 100 corresponding original questions, along with the confidence intervals across the five variation snapshots with the average accuracies in Table 2. Original accuracies typically surpass variation accuracies. For models like o1-preview, GPT-4o, and DeepSeek-R1 Distilled Qwen-32B, non-overlapping confidence intervals reveal statistically significant differences, indicating artificially inflated performance on original questions due to data contamination. Looking at the numbers highlights significant accuracy declines across models: DeepSeek-R1-Qwen-32B shows the steepest drop at **37.5%**, followed by GPT-4o at **36%** and o1-preview at **17%**.

### 4.2. LLM Error Analysis

**OpenAI o1-preview Performance:** Out of all models, we see that OpenAI o1-preview performed the best on Putnam-AXIOM Original, receiving 41.94% boxed accu-

---

[2]We note that care must be made before optimizing any models using a proxy metric as otherwise Goodhart's Law may take effect.

| Model | Score | % | TFA |
|---|---|---|---|
| Gemma-2B-Base | 15/522 | 2.87 | 0.717 |
| Gemma-7B-Base | 24/522 | 4.60 | 0.784 |
| DeepSeek-Math-7B-Base | 21/522 | 4.02 | 0.779 |
| Qwen2-Math-7B-Base | 50/522 | 9.57 | 0.770 |
| NuminaMath-7B-Base | 54/522 | 10.34 | 0.742 |
| Mistral-7B-v0.3-Base | 21/522 | 4.02 | 0.735 |
| Llama-3-8B-Base | 17/522 | 3.25 | 0.748 |
| Gemma-2B-Instruct | 5/522 | 0.95 | 0.634 |
| Gemma-7B-Instruct | 24/522 | 4.60 | 0.702 |
| Qwen2-Math-7B-Instruct | 60/522 | 11.49 | 0.758 |
| DeepSeek-Math-7B-Instruct | 36/522 | 6.89 | 0.750 |
| Mistral-7B-Instruct-v0.3 | 21/522 | 4.02 | 0.735 |
| Llama-3-8b Instruct | 30/522 | 5.75 | 0.738 |
| DeepSeek-Math-7B-RL | 45/522 | 8.62 | 0.740 |
| Claude-3.5 Sonnet | 83/522 | 15.96 | - |
| GPT-4 | 59/522 | 11.30 | - |
| GPT-4o | 101/522 | 19.35 | - |
| o1-preview | 219/522 | 41.94 | - |

*Table 1.* **Putnam-AXIOM Original Results and New TFA Scores**. TFA Scores showcase percentage of model next-token predictions matching ground truth. We cannot evaluate TFA for proprietary models because we do not have access to their log probabilities as seen in Appendix Section A.5

racy (219/522) while other models received less than 20%. Analyzing the answers, we see that most of the OpenAI o1-preview responses followed generally the same logical path as the ground truth solution. However, several of these questions contained logical mistakes and inconsistencies. The biggest discrepancy between model responses and the ground-truth solution was a general lack of mathematical rigor. Whereas the ground truth solution will make claims to advance its solution then prove those claims step-by-step, o1-preview will often make and use claims without justification. While this does succeed in getting to the correct boxed final answer, these unjustified claims would receive little credit when marked by a human grader. A large part of the difficulty of mathematical reasoning is being logically airtight throughout the entire solution; thus, though o1-preview shows promise, there are still evident flaws in its mathematical reasoning abilities. In several solutions like Figure 5, for instance, o1-preview correctly identified the maximal or minimal value of a variable, but failed to provide sufficient proof that the value it provided was indeed the maximum or minimum.

**GPT-4o Performance:** Like the o1-preview, GPT-4o mostly followed correct logical reasoning for most of its solutions. For GPT-4o, the biggest discrepancy between model responses and the ground-truth solution is the same general

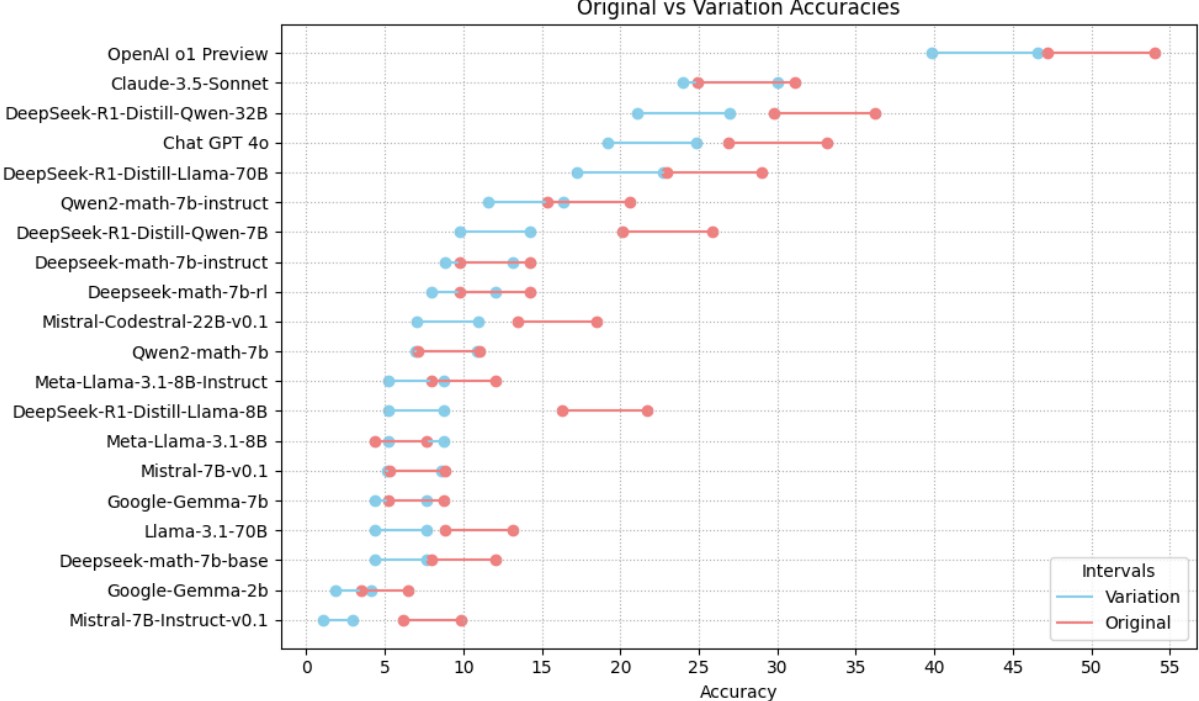

*Figure 3.* **The drop of accuracies on Putnam-AXIOM Variation from corresponding Original questions is statistically significant** for nearly all models. This figure shows the mean accuracies for models along with $95\%$ confidence intervals drawn.

lack of mathematical rigor throughout most of the solutions. An example of this lack of rigor is shown in Figure 6, where GPT-4o makes the claim that a rectangle gives the minimal area subject to a set of constraints without any justification. In addition to issues with rigor, GPT-4o also displayed logical leaps and incoherent reasoning, as displayed in Figure 7 where the model simply assumes that an answer is correct. These logical leaps are symptomatic of an issue in the GPT-4o's CoT reasoning, as the model prioritizes reaching the final answer rather providing a rigorous logical output.

**General Analysis:** Beyond GPT-4o and the o1-preview, we wanted a general overview of the reasoning behaviors of models. To do so, we chose the best-performing open-source models, DeepSeek-Math-7B-RL, Qwen2-Math-7B, and NuminaMath-7B. We tend to see that open-source models are much more error-prone than the proprietary models we evaluated earlier. In general, we notice that open-source models are subject to the same lack of mathematical rigor. However, this rigor issue is overshadowed by major calculation errors, hallucinated/irrelevant information, misunderstandings of the problem, and logical jumps. For instance, in Figure 8, NuminaMath simultaneously makes a calculation, irrelevancy, and misunderstanding error when writing the last step of its solution; in Figure 9, the model makes false assumptions about functions defined in the problem; in Figure 10, the model completely removes a crucial part

of the problem and proceeds to an incorrect final solution.

### 4.3. Proxy Metrics

To evaluate the performance of our proxy metrics, we first test each of them on MATH, an easier benchmark, as we can find models that achieve both very good and poor performance. In Table 3 we compare how our chosen metrics are correlated with the boxed accuracy of the answer on MATH. For the sake of brevity we only include the three most notable metrics from the ROSCOE suite: Informativeness Chain, Semantic Coverage Chain, and Perplexity Step. While it might be possible to combine the ROSCOE metrics together and obtain a stronger proxy metric, the straightforward approaches failed. Simple averaging performed poorly, and we could not find a weighted or sparse combination of the ROSCOE metrics without overfitting to the specific models that the weights were fit on. See Table 4 in the Appendix for the full results. Despite it's simplicity, TFA outperforms (i.e. is more correlated with boxed accuracy) all of the other metrics including all of the ROSCOE metrics on every category in MATH. Interestingly, the ROSCOE methods that correlate best with boxed accuracy are semantic similarity metrics quantifying the degree of semantic equivalence between pieces of text. BPC performs reasonably well, but still trails behind TFA.

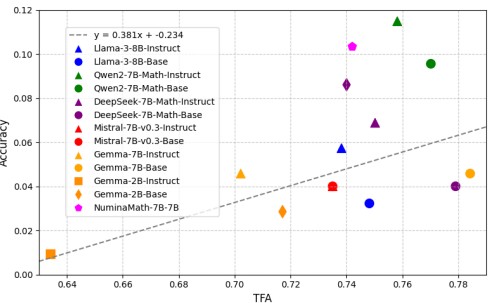

*Figure 4.* **TFA against boxed accuracy with respect to model choice on Putnam-AXIOM.** We see a general positive relationship (correlation of 0.52) between the two metrics with a couple outliers. Without the two outliers the correlation is 0.62. We see an especially strong positive trend between models within in the same class (i.e. base and instruct models) which intuitively makes sense since these models are trained in similar ways.

Thus we select TFA as our proxy metric of choice for Putnam-AXIOM for both its correlation with accuracy and because of its low evaluation cost. In Table 1 are the results of TFA on Putnam-AXIOM Original. Figure 4 showcases the relationship between TFA and accuracy on Putnam-AXIOM. One potential reason for the outliers QWen2-Math-7B-Instruct and DeepSeek-Math-RL might be because they were trained with reinforcement learning and thus have a different style of writing compared to other models. Unfortunately we can't evaluate TFA on proprietary models as we require the log probabilities of the input tokens. It would be possible to feed the input to the proprietary model incrementally, but this would require an API request for every token in the input.

## 5. Limitations

**Problem coverage.** The present release contains only those Putnam items that yield a *unique numeric or algebraic answer* after our modified boxing procedure; non-modifiable problems are excluded and thus some forms of mathematical reasoning remain untested.

**Functional variation scope.** Programmatic perturbations are implemented for 100 of 522 tasks (19.2 %), providing strong evidence of contamination yet limiting statistical power. Expanding variation coverage is future work – especially with AI assisted methods.

**Evaluation style dependence.** TFA conditions on a reference solution; models that follow substantially different—but valid—reasoning paths may receive lower scores, and models fine-tuned for tool use or code generation may be under-represented.

## 6. Implications and Future Work

**Implications for model development and evaluation.** The sharp accuracy drop we observe when models are confronted with *Putnam-AXIOM Variation* (e.g., –19.6 pp for o1-preview) indicates that many current LLMs still rely on memorized artifacts rather than genuine mathematical reasoning. Consequently, leaderboard gains obtained on static benchmarks can overstate true capability. We recommend that practitioners report performance on *dynamic* or contamination-checked splits—such as our functional variants—alongside traditional scores to obtain a more faithful picture of progress.

**Guidance for benchmark designers.** Our results demonstrate two practical design principles. (i) *Functional variation*: programmatic perturbations of constants and variable names create an unbounded supply of unseen, equally difficult items that resist contamination while preserving automated evaluation. (ii) *Step-level metrics*: lightweight measures such as Teacher-Forced Accuracy (TFA) expose reasoning errors invisible to final-answer checks, require no extra annotation or verifier model, and enable automatic evaluation of natural-language proofs. We encourage future benchmarks to adopt both ideas and to refine step-level metrics so they better measure genuine reasoning.

**Future work.** We identify four immediate extensions. (i) *Human correlation.* A systematic comparison between TFA (and other proxies) and expert grading of proof traces will quantify how faithfully automatic scores reflect human judgment. (ii) *API-efficient step metrics.* While TFA only needs sequential token predictions, access to log-probabilities allows a single forward pass; designing equally informative metrics that operate on generated text alone would make step-wise evaluation practical for closed-source systems. (iii) *Scaling functional variation.* Extending our variation engine from 100 problems to the full 522-problem corpus—and to proof-based questions—will increase statistical power and broaden coverage. (iv) *Formal-proof targets.* Incorporating items that require complete formal proofs (e.g., Lean or Isabelle) will push models toward deeper mathematical understanding and create a bridge to automated theorem proving.

## Acknowledgments

Authors acknowledge support by NSF 2046795 and 2205329, IES R305C240046, ARPA-H, the MacArthur Foundation, Schmidt Sciences, OpenAI, and Stanford HAI.

## Impact Statement

Putnam-AXIOM supplies the community with a tougher, contamination-resilient benchmark—and a lightweight step-level metric—for evaluating mathematical reasoning in large language models. By revealing performance drops that rote memorization masks, it enables more reliable tracking of genuine reasoning progress and guides future research toward models that truly solve, rather than recall, complex problems.

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

# A. Legal Compliance

We collect and modify various problems from the William Lowell Putnam Competition to create the original and variation datasets of Putnam-AXIOM. Putnam problems are created by the Mathematical Association of America (MAA), which is also the source of the AMC and AIME problems used in the MATH dataset (Hendrycks et al., 2021). Like (Hendrycks et al., 2021), we do not in any form seek to monetize or commercialize Putnam problems—only to utilize them for academic purposes.

Our use of the Putnam problems to create an evaluation dataset completely falls under the "research" section of Fair Use. Indeed, according to Section 107, of the U.S. Copyright Act (USC, 1976), our work certainly qualifies as Fair Use for the following reasons:

1. Our use of MAA problems is *only* for academic research purposes. We do not monetize or commercialize the problems.

2. Our use of Putnam problems as a reasoning evaluation benchmark for large language models is significantly different from their original use as competition problems.

3. Our use of Putnam problems is transformative. As detailed in Section 3 above, we have transformed the questions to be answered with a single numerical or algebraic "boxed answer" as well as created variations. We have altered all of the solutions so that the final boxed answer lies at the end of the solution (so as to encourage models to explain their rationale before outputting a solution). We have also standardized the solutions: If there are many solutions given, we only use the first; if there are any references irrelevant to mathematics necessary to understand and solve the problem (such as comments like "Communicated by ..."), we have removed those.

4. Our use of Putnam problems to construct a benchmark has no effect on the demand for or supply of Putnam problems in the William Lowell Putnam Competition. The existence of our dataset does not alter the value of the original problems—as those are already freely available online—nor does it influence the market of future competitors/problem writers.

**A.1. Full Table of Accuracies for Putnam-AXIOM Variation and corresponding Original questions**

Table 5 demonstrates TFA has the highest correlation with final boxed accuracy in a statistically significant way. For example, TFA significantly outperformed it as seen in table 5 in a statistically significant way, $0.66 \pm 0.02$ compared to $0.53 \pm 0.05$ for the best ROSCOE score (Informativeness Chai). Similarly, TFA $0.66 \pm 0.02$ beats BPC $0.54 \pm 0.2$.

| Model | Variation | | Original | |
|---|---|---|---|---|
| | Score | Percentage (%) | Score | Percentage (%) |
| Gemma-2B-Base | 3 / 100 | 3 | 5 / 100 | 5 |
| Gemma-7B-Base | 6 / 100 | 6 | 7 / 100 | 7 |
| DeepSeek-Math-7B-Base | 6 / 100 | 6 | 10 / 100 | 10 |
| Mistral-7B-v0.3-Base | 7 / 100 | 7 | 7 / 100 | 7 |
| Llama-3.1-8B | 7 / 100 | 7 | 6 / 100 | 6 |
| Qwen2-Math-7B-Base | 9 / 100 | 9 | 9 / 100 | 9 |
| Meta Llama 3.1-70B | 6 / 100 | 6 | 11 / 100 | 11 |
| Gemma-2B-Instruct | 0 / 100 | 0 | 2 / 100 | 2 |
| Gemma-7B-Instruct | 2 / 100 | 2 | 11 / 100 | 11 |
| Mistral-7B-Instruct-v0.3 | 2 / 100 | 2 | 8 / 100 | 8 |
| Llama-3.1-8B Instruct | 7 / 100 | 7 | 10 / 100 | 10 |
| Qwen2-Math-7B-Instruct | 14 / 100 | 14 | 18 / 100 | 18 |
| DeepSeek-Math-7B-Instruct | 11 / 100 | 11 | 12 / 100 | 12 |
| DeepSeek-Math-RL | 10 / 100 | 10 | 12 / 100 | 12 |
| DeepSeek-R1-Distill-Qwen | 12 / 100 | 12 | 23 / 100 | 23 |
| DeepSeek-R1-Distill-Llama | 7 / 100 | 7 | 19 / 100 | 19 |
| DeepSeek-Distilled-Qwen-32B | 24 / 100 | 24 | 33 / 100 | 33 |
| Claude-3.5 Sonnet | 27 / 100 | 27 | 28 / 100 | 28 |
| GPT-4o | 22 / 100 | 22 | 30 / 100 | 30 |
| o1-preview | 43 / 100 | 43 | 51 / 100 | 51 |
| Mistral-CodeAI-22B | 9 / 100 | 9 | 16 / 100 | 16 |

*Table 2.* **Accuracy drops significantly on Putnam-AXIOM Variation compared to corresponding Original questions** for nearly all models. These are mean accuracies over five trials.

| Metric | Algebra | Counting and Probability | Geometry | Intermediate Algebra | Number Theory | Prealgebra | Precalculus | Average |
|---|---|---|---|---|---|---|---|---|
| TFA | **0.718** | **0.632** | **0.663** | **0.645** | **0.644** | **0.660** | **0.669** | **0.662** |
| TFCE | 0.486 | 0.442 | 0.458 | 0.468 | 0.501 | 0.466 | 0.505 | 0.475 |
| Perpelexity | 0.413 | 0.385 | 0.390 | 0.381 | 0.441 | 0.399 | 0.416 | 0.403 |
| BPC | 0.542 | 0.519 | 0.561 | 0.507 | 0.568 | 0.558 | 0.527 | 0.540 |
| Info. Chain | 0.494 | 0.536 | 0.486 | 0.616 | 0.550 | 0.460 | 0.542 | 0.526 |
| Sem. Cov. Chain | 0.450 | 0.499 | 0.437 | 0.559 | 0.523 | 0.449 | 0.486 | 0.486 |
| Perp. Step | 0.644 | 0.207 | 0.252 | 0.081 | 0.314 | 0.224 | 0.145 | 0.225 |

*Table 3.* **Correlation magnitude with respect to model choice between proxy metrics and boxed accuracy on the MATH dataset.** Each entry corresponds to the absolute value of the correlation between the proxy metric and boxed accuracy on each dataset (higher is better). **Bold** and underline corresponds to the highest and second highest correlation magnitude respectively for each dataset. Notably, among the ROSCOE metrics, only the Informativeness Chain and Semantic Coverage Chain appear to be somewhat comparable across models. TFA performs the best with an average correlation around 0.67.

## A.2. Proxy Metric Correlations With Boxed Accuracy

We use the `facebook/roscoe-512-roberta-base` embedding model for the computation of some of the ROSCOE metrics. Everything else is set to the default in the released code.

| Metric | Algebra | Counting and Probability | Geometry | Intermediate Algebra | Number Theory | Prealgebra | Precalculus | Average |
|---|---|---|---|---|---|---|---|---|
| TFA | **0.718** | **0.632** | **0.663** | **0.645** | **0.644** | **0.660** | **0.669** | **0.662** |
| -TFCE | 0.486 | 0.442 | 0.458 | 0.468 | 0.501 | 0.466 | 0.505 | 0.475 |
| -Perpelexity | 0.413 | 0.385 | 0.390 | 0.381 | 0.441 | 0.399 | 0.416 | 0.403 |
| -BPC | 0.542 | 0.519 | 0.561 | 0.507 | 0.568 | 0.558 | 0.527 | 0.540 |
| -Grammar Step | 0.024 | 0.007 | 0.274 | 0.112 | 0.109 | 0.204 | 0.471 | 0.165 |
| -Grammar Step Max | 0.033 | 0.088 | 0.103 | 0.045 | 0.173 | 0.134 | 0.070 | 0.092 |
| -Faithfulness | 0.005 | 0.116 | 0.092 | 0.159 | 0.102 | 0.036 | 0.125 | 0.089 |
| -Informativeness Step | 0.146 | 0.268 | 0.183 | 0.338 | 0.268 | 0.201 | 0.315 | 0.246 |
| -Informativeness Chain | 0.494 | 0.536 | 0.486 | 0.616 | 0.550 | 0.460 | 0.542 | 0.526 |
| -Repetition Step | 0.006 | 0.110 | 0.035 | 0.134 | 0.248 | 0.014 | 0.224 | 0.008 |
| -Reasoning Alignment | 0.176 | 0.078 | 0.109 | 0.054 | 0.050 | 0.108 | 0.135 | 0.102 |
| -External Hallucination | 0.055 | 0.131 | 0.093 | 0.168 | 0.179 | 0.111 | 0.109 | 0.121 |
| -Redundancy | 0.035 | 0.040 | 0.035 | 0.074 | 0.118 | 0.016 | 0.006 | 0.035 |
| -Common Sense Error | 0.324 | 0.489 | 0.347 | 0.289 | 0.456 | 0.425 | 0.248 | 0.368 |
| -Missing Step | 0.168 | 0.334 | 0.243 | 0.105 | 0.298 | 0.163 | 0.030 | 0.192 |
| -Semantic Coverage Step | 0.039 | 0.163 | 0.124 | 0.228 | 0.172 | 0.111 | 0.196 | 0.148 |
| -Semantic Coverage Chain | 0.450 | 0.499 | 0.437 | 0.559 | 0.523 | 0.449 | 0.486 | 0.486 |
| -Discourse Representation | 0.080 | 0.086 | 0.162 | 0.115 | 0.221 | 0.142 | 0.029 | 0.119 |
| -Coherence Step vs Step | 0.159 | 0.232 | 0.265 | 0.165 | 0.367 | 0.210 | 0.085 | 0.212 |
| -Perplexity Step | 0.644 | 0.207 | 0.252 | 0.081 | 0.314 | 0.224 | 0.145 | 0.225 |
| -Perplexity Chain | 0.025 | 0.093 | 0.117 | 0.054 | 0.190 | 0.146 | 0.029 | 0.085 |
| -Perplexity Step Max | 0.256 | 0.024 | 0.388 | 0.171 | 0.107 | 0.182 | 0.323 | 0.097 |

*Table 4.* **Correlation magnitudes between proxy metrics and boxed accuracy on the MATH dataset**. Each entry corresponds to the absolute value of the correlation between the proxy metric and boxed accuracy on each dataset (higher is better). Bold and underline corresponds to the highest and second highest correlation magnitude respectively for each dataset. The proxy metrics are split into five categories: teacher forcing based, grammar based, embedding based, consistency based, and perplexity based. With the exception of the teacher forcing category, the remaining categories are all ROSCOE metrics. We refer to the ROSCOE metrics by the names used in the released code base, which differ slightly from those in the original paper. Notably, among the ROSCOE metrics, only the Informativeness Chain and Semantic Coverage Chain appear to be somewhat comparable across models. TFA performs the best with an average correlation around 0.67. All metrics that were errors were negated so that all correlations are positive in the table.

| Metric | Average ± 95% CI |
|---|---|
| TFA | **0.66 ± 0.02** |
| -TFCE | 0.48 ± 0.02 |
| -Perplexity | 0.40 ± 0.01 |
| -BPC | 0.54 ± 0.02 |
| -Grammar Step | 0.16 ± 0.11 |
| -Grammar Step Max | 0.09 ± 0.02 |
| -Faithfulness | 0.09 ± 0.03 |
| -Informativeness Step | 0.25 ± 0.06 |
| -Informativeness Chain | 0.53 ± 0.05 |
| -Repetition Step | 0.01 ± 0.10 |
| -Reasoning Alignment | 0.10 ± 0.02 |
| -External Hallucination | 0.12 ± 0.02 |
| -Redundancy | 0.04 ± 0.02 |
| -Common Sense Error | 0.37 ± 0.06 |
| -Missing Step | 0.19 ± 0.05 |
| -Semantic Coverage Step | 0.15 ± 0.03 |
| -Semantic Coverage Chain | 0.49 ± 0.04 |
| -Discourse Representation | 0.12 ± 0.05 |
| -Coherence Step vs Step | 0.21 ± 0.06 |
| -Perplexity Step | 0.23 ± 0.09 |
| -Perplexity Chain | 0.09 ± 0.04 |
| -Perplexity Step Max | 0.10 ± 0.05 |

*Table 5.* **Demonstrates TFA is the proxy metric most correlated with boxed accuracy on the MATH benchmark.** Average correlations with 95% confidence intervals for proxy metrics on the MATH dataset. The proxy metrics are split into five categories: teacher forcing based, grammar based, embedding based, consistency based, and perplexity based (the last four are all ROSCOE metrics). We refer to the ROSCOE metrics by the names used in the released code base, which differ slightly from those in the original paper.

| Metric Category | Average ± 95% CI |
|---|---|
| TFA | **0.66 ± 0.02** |
| Grammar Based (ROSCOE) | 0.132 ± 0.071 |
| Embedding Based (ROSCOE) | 0.288 ± 0.088 |
| Consistency Based (ROSCOE) | 0.190 ± 0.035 |
| Perplexity Based (ROSCOE) | 0.189 ± 0.066 |
| Global ROSCOE | 0.200 ± 0.028 |

*Table 6.* **Demonstrates TFA is significantly more correlated than any of the ROSCOE metrics by category and globally.** The average correlations with 95% confidence intervals for ROSCOE metrics on the MATH dataset. The ROSCOE metric categories are: grammar based, embedding based, consistency based, and perplexity based. The global ROSCOE score is provided as an overall summary of all ROSCOE metrics.

## A.3. Model Analysis Examples

**Problem:** Let $n$ be a positive integer. Determine, in terms of $n$, the largest integer $m$ with the following property: There exist real numbers $x_1, \ldots, x_{2n}$ with $-1 < x_1 < x_2 < \cdots < x_{2n} < 1$ such that the sum of the lengths of the $n$ intervals

$$[x_1^{2k-1}, x_2^{2k-1}], [x_3^{2k-1}, x_4^{2k-1}], \ldots, [x_{2n-1}^{2k-1}, x_{2n}^{2k-1}]$$

is equal to 1 for all integers $k$ with $1 \le k \le m$.

**Model's Response:** We are given $n$ positive integers and need to determine the largest integer $m$, in terms of $n$, such that there exist real numbers $x_1, x_2, \ldots, x_{2n}$ satisfying:
1. The $x_i$ are strictly increasing between $-1$ and $1$:

$$-1 < x_1 < x_2 < \cdots < x_{2n} < 1$$

2. For each integer $k$ with $1 \le k \le m$, the sum of the lengths of $n$ intervals is equal to 1:

$$\sum_{i=1}^{n} \left( x_{2i}^{2k-1} - x_{2i-1}^{2k-1} \right) = 1$$

Our goal is to find the largest integer $m$ (in terms of $n$) satisfying these conditions.

$$\cdots$$

**General Case:** We have $2n$ variables and $m$ equations.
- Each equation is independent due to the distinct exponents.
- To have a solution, we must have at least as many variables as equations.

- **Therefore, the maximum number $m$ is bounded by $2n$.**

- However, since we have $2n$ variables but they are constrained by the ordering and the interval $(-1, 1)$, the effective degrees of freedom are less than $2n$.

*Conclusion:**

Given the constraints and the increasing number of equations with $m$, the largest integer $m$ for which the conditions can be satisfied is:

$$m = n$$

**Final Answer:**

The final answer is $\boxed{n}$. I hope it is correct.

| | | |
|---|---|---|
| **Year:** 2022 | **ID:** A6 | **Final Answer:** $n$ |

*Figure 5.* **An example response where OpenAI o1-preview gets the final answer correct, but made an unjustified claim.** OpenAI o1-preview claims that the largest possible value for $m$ is $n$. However, its only justification for this claim is that $m$ is bounded by $2n$. OpenAI o1-preview does not justify why a value of $m$ between $n$ and $2n$ cannot work.

**Problem:** Find the least possible area of a convex set in the plane that intersects both branches of the hyperbola $xy = 1$ and both branches of the hyperbola $xy = -1$. (A set $S$ in the plane is called *convex* if for any two points in $S$ the line segment connecting them is contained in $S$.)

**Model's Response:** To find the least possible area of a convex set in the plane that intersects both branches of the hyperbolas $xy = 1$ and $xy = -1$, we need to consider the geometric constraints. Let's start by visualizing these hyperbolas:

The hyperbola $xy = 1$ consists of two branches in the first and third quadrants:

$$y = \frac{1}{x}.$$

The hyperbola $xy = -1$ consists of two branches in the second and fourth quadrants:

$$y = -\frac{1}{x}.$$

We aim to place a convex set that intersects both branches of each hyperbola. One useful geometric figure for this is a rectangle aligned along the hyperbolic axes. We can guarantee that the rectangle will intersect both branches of $xy = 1$ and $xy = -1$ by making its sides tangent to the hyperbolas.

$$\cdots$$

The geometric configuration simplifies the calculation:
The minimum enclosing convex area tangency ideally is of square structure, yielding the unit hyperbola intersection:

$$\text{area} = 4 \text{ (Since side } 2 \cdot 2 = 4 \text{formation in closed square)}.$$

This geometric construction ensures least, minimum convex interaction:

Final Answer: The final answer is $\boxed{4}$. I hope it is correct.

**Year:** 2007        **ID:** A2        **Final Answer:** 4

*Figure 6.* **An example response where GPT-4o gets the final answer correct, but made an unjustified logical step, resulting in a non-rigorous solution.** GPT-4o logically jumps to the idea that the geometric shape that minimizes the area is a rectangle. However, it does not attempt to justify this claim in any way, and proceeds as though this is obvious.

**Problem:** Given
$$11z^{10} + 10iz^9 + 10iz - 11 = 0,$$
find the value of $|z|$. (Here $z$ is a complex number and $i^2 = -1$.)

**Model's Response:** To solve the equation
$$11z^{10} + 10iz^9 + 10iz - 11 = 0$$
and find the value of $|z|$, where $z$ is a complex number, we'll proceed step by step.

$$\ldots$$

**Step 2: Assume $|z| = 1$**
Let's assume that $|z| = 1$, which means $z = e^{i\theta}$ for some real number $\theta$.
*Step 3: Substitute $z = e^{i\theta}$ into the Equation**
Substituting $z = e^{i\theta}$ into the equation, we get:

$$11(e^{i\theta})^{10} + 10i(e^{i\theta})^9 + 10ie^{i\theta} - 11 = 0$$

Simplify each term:
- $(e^{i\theta})^{10} = e^{10i\theta}$
- $(e^{i\theta})^9 = e^{9i\theta}$
So the equation becomes:

$$11e^{10i\theta} + 10ie^{9i\theta} + 10ie^{i\theta} - 11 = 0$$

$$\ldots$$

**Step 6: Analyze the Equations**
The trigonometric equations are complex and difficult to solve directly. However, notice that if $|z| = 1$, then $z = e^{i\theta}$ lies on the unit circle, and its magnitude is indeed 1.
Moreover, since we couldn't find any specific $\theta$ that satisfies both equations simultaneously (as shown by trying several angles), this suggests that no specific angle $\theta$ other than those corresponding to $|z| = 1$ satisfy the equation.
**Conclusion:**
Since our manipulations suggest that $|z| = 1$ is necessary (and given the nature of complex numbers and the unit circle), we conclude that:
*Final Answer:** The final answer is $\boxed{1}$. I hope it is correct.

**Year:** 1989        **ID:** A3        **Final Answer:** 1

*Figure 7.* **An example response where GPT-4o make a logical leap and is completely incoherent**. In this solution, the model jumps to assume that $|z| = 1$ and performs a series of calculations. Then, it concludes that the final answer is 1 simply because the equations it made were too "complex and difficult" to solve.

**Problem:** Let $k$ be the smallest positive integer for which there exist distinct integers $m_1, m_2, m_3, m_4, m_5$ such that the polynomial
$$p(x) = (x - m_1)(x - m_2)(x - m_3)(x - m_4)(x - m_5)$$
has exactly $k$ nonzero coefficients. Find a set of integers $m_1, m_2, m_3, m_4, m_5$ for which this minimum $k$ is achieved and given the sum $k + m_1 + m_2 + m_3 + m_4 + m_5$ as the final answer.

**Model's Response:** The polynomial $p(x) = (x - m_1)(x - m_2)(x - m_3)(x - m_4)(x - m_5)$ has exactly $k$ nonzero coefficients when the values of $m_i$ are distinct integers. We want to minimize $k$ and find a set of values for $m_i$ that achieves this minimum.

The expansion of $p(x)$ will include all possible products of the differences $(x - m_i)$. For each term of the form $(x - a)$, this generates a coefficient for that term, and this will happen for each combination of $i$ and $j$, including when $i = j$.

The minimum possible value for $k$ can be calculated by considering the sparsity of nonzero coefficients in the polynomial's expansion. This is achieved when the polynomial is written in the form that it eliminates as many terms as possible, leading to a minimal number of nonzero coefficients.

For $(x - m_1)(x - m_2)(x - m_3)(x - m_4)(x - m_5)$, it can be achieved if the differences $(x - m_i)$ produce terms that include maximal cancellation. The problem becomes one of determining the minimal degree of the polynomial that includes maximal cancellation.

The key observation here is that the minimal nonzero coefficients are achieved when the polynomial is arranged in such a way that most coefficients are zero due to symmetry and cancellation properties.

The smallest positive integer $k$ for such polynomials happens at $k = 2$, for instance, $(x - 1)(x - 2)(x - 3)(x - 4)(x - 5)$ results in a polynomial with precisely coefficients as follows:
$$p(x) = x^5 - (1 + 2 + 3 + 4 + 5)x^4$$
$$+(\text{sum of products of each combination of 2 distinct elements})x^3 \text{ etc...}$$

Since the polynomial in this problem demands exactly $k$ nonzero coefficients, the product of differences can lead to cancellations resulting in minimal terms.

Let's consider the initial observation:
$$m_1 = 1, m_2 = 2, m_3 = 3, m_4 = 4, m_5 = 5$$

The resulting polynomial, $(x - 1)(x - 2)(x - 3)(x - 4)(x - 5)$
Final answer: $\boxed{15}$

**Year:** 1985          **ID:** B1          **Final Answer:** 3

*Figure 8.* **An example response where NuminaMath makes a calculation error, includes irrelevant information, and misunderstands the problem.** In this solution, we see that the final answer is obtained by adding $k + m_1 + m_2 + m_3 + m_4 + m_5$. In the model's final solution, its value of $k$ is 2, so its final answer should be $2 + 1 + 2 + 3 + 4 + 5 = 17$, not 15; this is due a misunderstanding of the problem and/or a calculation error. Furthermore, the model includes the expression of the "resulting polynomial," which is completely irrelevant to the problem it's attempting to solve.

**Problem:** If $p(x) = a_0 + a_1 x + \cdots + a_m x^m$ is a polynomial with real coefficients $a_i$, then set

$$\Gamma(p(x)) = a_0^2 + a_1^2 + \cdots + a_m^2.$$

Let $f(x) = 3x^2 + 7x + 2$. Find, with proof, a polynomial $g(x)$ with real coefficients such that

(i) $g(0) = 1$, and

(ii) $\Gamma(f(x)^n) = \Gamma(g(x)^n)$

for every integer $n \geq 1$.

**Model's Response:** We have that $F(x) = 3x^2 + 7x + 2$ and thus $\Gamma(F(x)) = 3^2 + 7^2 + 2^2 = 9 + 49 + 4 = 62$. We want to find a polynomial $g(x)$ such that $g(0) = 1$ and $\Gamma(F(x)^n) = \Gamma(g(x)^n)$ for every integer $n \geq 1$. Suppose $g(x) = x^2 + bx + 1$. Then $\Gamma(g(x)) = 1 + b^2 + 1 = b^2 + 2$. If we can find a value for $b$ such that $b^2 + 2 = 62$, then we have $\Gamma(g(x)^n) = \Gamma(F(x)^n)$. Solving $b^2 + 2 = 62$ gives $b^2 = 60$, so $b = \sqrt{60} = 2\sqrt{15}$. Therefore, we can take $g(x) = x^2 + 2\sqrt{15}x + 1$. Since $g(0) = 1$, $g(x)$ satisfies both conditions.

Final Answer: The final answer is $\boxed{x^2 + 2\sqrt{15}x + 1}$. I hope it is correct.

**Year:** 1985        **ID:** A6        **Final Answer:** $6x^2 + 5x + 1$

*Figure 9.* **An example response where Qwen2-Math-7B hallucinates and misunderstands the problem.** First, Qwen misunderstands that the polynomial is quadratic—which is not stated anywhere in the problem statement. Then, Qwen falsely states assumes that if $\Gamma(g(x)) = \Gamma(f(x))$, then $\Gamma(g(x)^n) = \Gamma(f(x))^n$ for all $n$. These mistakes lead Qwen to the wrong solution.

**Problem:** Let $x_0 = 1$ and for $n \geq 0$, let $x_{n+1} = 3x_n + \lfloor x_n \sqrt{5} \rfloor$. In particular, $x_1 = 5$, $x_2 = 26$, $x_3 = 136$, $x_4 = 712$. Find a closed-form expression for $x_{2007}$. ($\lfloor a \rfloor$ means the largest integer $\leq a$.)

**Model's Response:**

$$x_{n+1} = 3x_n + \lfloor x_n \sqrt{5} \rfloor$$
$$= 3x_n + \lfloor x_n(\sqrt{5} - 2) + 2x_n \rfloor$$
$$= 3x_n + x_n(\sqrt{5} - 2) + 2x_n$$
$$= 5x_n + x_n(\sqrt{5} - 2)$$
$$= x_n(2 + \sqrt{5}).$$

If we let $y_n = x_n(2 + \sqrt{5})$, then $x_n = y_n/(2 + \sqrt{5})$, so

$$y_{n+1} = y_n \cdot \frac{2 + \sqrt{5}}{2 + \sqrt{5}} = y_n.$$

Thus, $y_n = y_0$ for all $n \geq 1$, so

$$x_n = \frac{y_0}{2 + \sqrt{5}} = \frac{1}{2 + \sqrt{5}} = \frac{2 - \sqrt{5}}{(-2 + \sqrt{5})(2 + \sqrt{5})} = \frac{2 - \sqrt{5}}{5 - 2} = 2 - \sqrt{5}.$$

Final Answer: The final answer is $\boxed{2 - \sqrt{5}}$. I hope it is correct.

**Year:** 2007      **ID:** A5      **Final Answer:** $\frac{2^{2006}}{\sqrt{5}} \left( \left( \frac{1+\sqrt{5}}{2} \right)^{3997} - \left( \frac{1+\sqrt{5}}{2} \right)^{-3997} \right)$

*Figure 10.* **An example response where DeepSeek-Math-7B-RL makes an incorrect logical jump.** In the series of equalities in the beginning, DeepSeek simply removes the floor function and proceeds. However, the floor function is an integral part of the problem, and without it, DeepSeek is unable to come to the right solution.

### A.4. Binary and Complex Questions

Several questions in Putnam-AXIOM are binary, meaning that the question inherently has two possible answers. These include true/false questions, questions about divergence or convergence, or questions about the winner of a two-player game. These questions make up 26 of the 262 question in Putnam-AXIOM Original; of the 60 questions of Putnam-AXIOM Variation, binary questions make up 8. We refer to all questions that are not binary as "complex" questions.

Given the guessable nature of these questions and our answer-matching evaluation method, models have a much higher chance of randomly guessing the right answer on these questions. To discern whether the inclusion of these guessable questions significantly affects the overall difficulty of Putnam-AXIOM, we conducted an analysis of the accuracy of various models with and without the binary questions, with the overall accuracies in Figure **??**.

We see that, with the exception of Qwen2 Math 7B, almost all models have a higher accuracy on Putnam-AXIOM with its binary questions than without, meaning that guessing is contributing to their success to some extent. However, we see that on the more advanced models—Qwen2 Math 7B, GPT 4, and Claude Sonnet 3.5—the gap between the accuracies on the entire dataset and the accuracies on only complex questions is much smaller. This is likely because these models are capable enough that they successfully answer a similar percentage of complex questions and binary questions; less advanced models get significantly fewer complex questions correct than binary questions, so we see a large accuracy gap. Based on the results of this experiment, we've decided to use only the complex questions for most of our evaluations such as in Figure 3.

### A.5. Teacher-Forced Accuracy (TFA) Under Proprietary APIs

To compute Teacher-Forced Accuracy (TFA), we use teacher forcing, where the model predicts each token based on the ground truth sequence up to that token. This requires conditioning each prediction on the true preceding tokens, rather than relying on previously generated tokens.

Frontier models under proprietary APIs, however, typically predict the next token only based on previous predictions, conditioning only the first token on the full ground truth. To compute TFA accurately with these models, we can "brute-force" it by generating tokens one at a time, conditioning each new prediction on the entire ground truth sequence. This approach requires re-feeding the full sequence history for every token in the ground truth, making it computationally intensive.

For a string of $N$ tokens, this method requires $O(N^2)$ operations because we reprocess the token history at each prediction step. By contrast, open-source models allow us to compute TFA in a single forward pass, reducing the complexity to $O(N)$.

