# OpenReview forum: "Putnam-AXIOM: A Functional & Static Benchmark for Measuring Higher Level Mathematical Reasoning in LLMs"
_ICML.cc/2025/Conference — ICML 2025 poster_

### Official Review · Reviewer_Vq1J · 2025-03-10

**Overall Recommendation:** 3

**Summary:**

This paper presents Putnam-AXIOM, a benchmark of 522 problems from the Putnam competition along with their ground truth solutions. The paper also proposes some manual modification to the original problems to make evaluation less ambiguous, as well as manual variable and constant substitutions for 100 problems (Putnam-AXIOM Variation). The paper evaluates both open-source and proprietary LLMs on the original and varied problems and show that LLMs achieve higher accuracy on the original problems, suggesting data contamination problem exists.

**Claims And Evidence:**

Some claims are inaccurate or exaggerating:

1. > On Line 103-105 of Related Work: "Our Putnam-AXIOM dataset addresses these limitations by offering challenging Putnam problems with fully-written solutions and easily evaluable answers."

I am under the impression that Putnam-AXIOM did not address the costly human evaluation issue for symbolic and proof-based questions. Instead, it tries to avoid this problem by only keeping the problems that are not proof-based.

2. > On Line 323-324 of Section 4.1:  Looking at the numbers highlights significant accuracy declines across models: DeepSeek-R1-Qwen-32B shows the steepest drop at 37.5%, followed by GPT-4o at 36% and o1-preview at 17%.

These numbers are misleading and it seems that the authors are subtracting the min of the lower accuracy type from the max of the higher accuracy type. From Table 2, it is clear that the accuracy drop is much less than what is reported in the main text.

**Essential References Not Discussed:**

N/A.

**Experimental Designs Or Analyses:**

The experimental designs and analyses are generally sound.

**Methods And Evaluation Criteria:**

The methods and evaluation criteria are generally sound.

**Other Comments Or Suggestions:**

N/A.

**Other Strengths And Weaknesses:**

**Strengths**
1. The paper motivates the problem well and provides a comprehensive evaluation of the benchmark with many models with different tiers of capabilities. The analysis also makes sense.
2. The modified boxing approach is very interesting and I believe this can be a promising way to sanitize benchmarks.

**Weaknesses**
1. The approach to create modified boxing and variation dataset especially the constant change seems to be largely a manual effort. The author also mentions that not all problems are suitable for such variation. This makes the approach less generally applicable and less novel.

2. Although TFA seems to correlate well with boxed accuracy, from Table 1 and Figure 4, it is clear that it cannot be used as a proxy for model selection. That is, the model with the highest TFA intuitively is not a strong model (Gemma-7B-Base) and boxed accuracy correlates better with general perception. This makes the TFA less appealing for practical use.

3. The paper has many presentation flaw that significantly affects understanding of the results. See questions below.

Overall, I believe the main novelty of the paper is on the strategies used for creation of variation dataset and alternative evaluation metric. However, the strategies appear to be manual (not very scalable without introducing errors), and the evaluation metric seems to be noisy.

**Questions For Authors:**

1. The below sentence is confusing and seems to come from nowhere. Can the authors explain what this means. Also does this modification affect the correlation performance of TFA?

> . Further, we standardized the placement of boxed answers by relocating them to the end of each solution string to minimize unintended emergent behaviors leading to evaluations that are less “harsh” or prone to penalizing the model for formatting deviations rather than actual comprehension.

2. In Table 1, NuminaMath-7B-Base has accuracy of 10.34%. However, in Figure 4, it is shown to be less than 5% with the pentagon symbol. Can the authors explain the inconsistency?

3. PutnamBench has 640 problems whereas this dataset has 522. What is causing the discrepancy and is it because proof-based problems were excluded?

4. In Figure 3, it seems that the original accuracy is generally lower than the variation accuracy. However, the text suggests original accuracy is higher. Am I missing something?


## Update After Rebuttal
After reading the author's rebuttal and the other reviews, I have decided to increase my rating. As noted by several reviewers, there are still issues with the writing and presentation, including errors and occasional misrepresentations. However, I believe that the paper's contributions outweigh these weaknesses.

**Relation To Broader Scientific Literature:**

The finds of the paper is interesting. However, the dataset proposed here has limited novelty due to existing work on PutnamBench.

**Theoretical Claims:**

N/A.

---

> ### Author Rebuttal · Authors · 2025-04-01
>
> We sincerely thank the reviewer for their thoughtful and detailed response. We deeply appreciate the time and effort you invested in providing such valuable feedback.
>
> > I am under the impression … not proof-based.
>
> We'd like to clarify that Putnam-AXIOM explicitly addresses the challenge of evaluating symbolic and proof-based questions through several carefully designed methodological choices:
>
> We retained numerous proof-based questions by applying our Modified Boxing technique (detailed in Section 3.1), which transforms these questions to produce unambiguous, final answers while preserving their mathematical complexity. This approach allows for automatic evaluation without sacrificing the deep reasoning required in the original problems.
>
> Our methodology supports a wide range of answer formats through our equivalence function, similar to the approach used in MATH [1]. For Putnam problems that involve complicated final answers, we added minimal next steps that maintain "the same mathematical insights and problem-solving approaches required by the original problems" while ensuring solutions converge to single, automatically evaluable answers.
>
> We deliberately excluded binary-answer questions (~10% of Putnam problems) to eliminate success through random guessing.
>
>
> [1] Hendrycks, D., Burns, C., Kadavath, S., Arora, A., Basart, S., Tang, E., ... & Steinhardt, J. (2021). Measuring mathematical problem solving with the math dataset. arXiv preprint arXiv:2103.03874.
>
> >These numbers are misleading … main text.
>
>
> Thank you for pointing out that mistake, the corrected values should be that “DeepSeek-R1-Qwen-32B shows the steepest drop at 27.3%, followed by GPT-4o at 26.7% and o1-preview at 15.6%." This correction will be reflected in the final version of our paper.
>
> However, we emphasize that our variations are highly effective at mitigating contamination. During rebuttal, we fine-tuned DeepSeek-R1 Distill-Qwen-7B (chosen as the best-performing model in its size class) on the full Putnam-AXIOM Original dataset for 5 epochs using full-parameter supervised fine-tuning (SFT) intended to mimic or exaggerate the effects of data contamination. The baseline model achieved 12% accuracy on variations versus 23% accuracy on corresponding originals. After fine-tuning, accuracy increased to 33% on variations versus 80% on corresponding originals. This demonstrates that while contamination allows near-saturation on original problems (+57% accuracy), variations remain challenging (+21% accuracy), confirming their effectiveness against contamination. We plan to extend these experiments to additional models.
>
> However, the dataset proposed here has limited novelty due to existing work on PutnamBench.
> Thank you for pointing out this possible confusion. As mentioned in our introduction, Putnam-AXIOM focuses on natural-language problems with final answer verification, functional variations to address contamination, and proxy metrics for reasoning. PutnamBench translates Putnam questions into formal statements for theorem-proving languages. Our datasets serve distinct purposes despite being sourced from the same competition.
>
> > The approach to create … less novel.
>
> Thank you for pointing out this concern. We agree that manual effort is required to create variations. While we created a script using GPT-4o to assist in variation creation, significant manual verification is still required. We note that most difficult benchmarks (like MATH) also require substantial manual effort.
>
> > Although TFA seems … practical use.
> Thank you for pointing this out. We acknowledge that boxed accuracy correlates better with general perception. However, per Goodhart's Law, optimizing too much for any proxy diminishes its value. We aim to create a fast, inexpensive proxy for reasoning, not necessarily the optimal metric.
>
> The below sentence is confusing and seems to come from nowhere … actual comprehension.
> We standardized placement of boxed answers at the end of solution strings based on findings from Chain-of-Thought prompting, where models provide more coherent solutions when reasoning through the entire problem before giving the final answer. For TFA, this means models are more penalized for formatting deviations and less penalized for solutions with reasoning similar to benchmark solutions.
>
> >PutnamBench has 640 problems … were excluded?
>
> This discrepancy is due to the fact that Putnam-AXIOM ultimately aims to serve a different purpose than PutnamBench: we selected problems from the Putnam based on their ability to contribute a final, boxed answer for our automatic evaluation process. PutnamBench also pruned ~60% of Putnam problems. We excluded binary-answer questions but included many proof-based questions by rewording them to produce evaluable final answers while preserving difficulty.
>
> >In Figure 3 … missing something?
>
> Thank you for pointing this out. The labels for Figure 3 were indeed mistakenly swapped. We will correct this in the final version.

---

### Official Review · Reviewer_W8M8 · 2025-03-10

**Overall Recommendation:** 3

**Summary:**

This paper introduces a new mathematical benchmark made of 522 questions from the William Lowell Putnam math competition from 1938 to 2023, among which 100 can be infinitely modified by changing variable names (100 of them) and constant values (37/100 of them) – called the Variation split.
This paper also presents teacher-forced accuracy (TFA) as an additional evaluation metric to give a more complete assessment of LLMs’ reasoning abilities.

Experiments show that current state-of-the-art models struggle to achieve good performance with OpenAI o1-preview achieving ±42% accuracy. Experiments also show that models evaluated on the Variation split are weaker than when evaluated on the corresponding 100 original questions (suggesting some data was present during pre-training).

## update after rebuttal

**Claims And Evidence:**

Claims are somewhat supported by evidence.

(1) The claim that TFA provides a more complete assessment of LLM’s reasoning abilities compared to final answer accuracy makes sense intuitively but is not supported by strong evidence (see explanation in Experimental Designs Or Analyses section).

(2) The claim that changing variable names, constant values, and turn of phrase will tackle data contamination is not convincing (see explanation in Other Strengths And Weaknesses section).

**Essential References Not Discussed:**

Nothing essential is missing.

**Experimental Designs Or Analyses:**

1. The motivation for using pseudo metrics like Teacher Force & ROSCOE is to evaluate the models at their reasoning skills because the final \boxed{} answer may not represent this well enough. However when evaluating these metrics, correlation with the final answer accuracy is used to select the top pseudo metric. This is not the correct approach as you already have a metric that will correlate 100% of the time with accuracy, it is accuracy itself. The whole point of using proxy metrics is to evaluate **something else** than accuracy, i.e. the reasoning chain logic. The correlation between these pseudo metrics should be evaluated against human labels or a strong LLM as a judge such as GPT4o, not against accuracy.

2. The numbers in Table 1 do not match the scatter plot in Figure 4. For instance:
- Gemma-7B-Base in Table 1: (0.046 acc, **0.784 TFA**) – in Figure 4: (±0.043 acc, **<0.74 TFA**)
- DeepSeek-Math-7B-Base in Table 1: (**0.0402 acc**, 0.779 TFA) – in Figure 4: (±**0.06 acc**, ±0.772 TFA)
- Qwen2-Math-7B-Base in Table 1: (**0.0957 acc**, 0.770 TFA) – in figure 4: (**<0.06 acc**, ±0.765 TFA)

In particular, the numbers in Table 1 do not seem to correlate as strongly as in the figure 4.

3. Section 4.3 says that the “_chosen metrics_” are reported in Table 3 (in the appendix) but Table 4 (in the appendix) shows that ‘’Common Sense Error’’ correlates more (0.368) than the chosen ‘’Perplexity Step’’ (0.225). Why was “perplexity Step” chosen instead of ‘’Common Sense Error’’?
In addition, what does it mean to be “chosen” for a pseudo metric if anyway at the end of section 4.3 only the top one (TFA) is “_selected_” to evaluate models (table 1 & figure 4).

4. Experimental results should report model’s performance on some of the most interesting proxy metrics. Note that “interesting” here should not mean high correlation with final answer accuracy, but rather a high correlation with human or LLM judgment of the reasoning chain logic. Intuitively, “Common Sense Error”, “Hallucination”, and “Repetition” or “Redundancy” would be things to avoid in a reasoning chain, and thus point towards interesting pseudo metrics (to be validated by llm or human as a judge experiments).

**Methods And Evaluation Criteria:**

Pseudo metrics (teacher forced, and ROSCOE) are evaluated based on their correlation with final answer accuracy. They should be evaluated based on other metrics that directly inspect the reasoning ability of LLMs (either human or LLM as a judge labels).

**Other Comments Or Suggestions:**

The caption in Figure 3 may be inverted: right now it shows that the accuracies on the Original questions are **weaker** than on the Variation split.

**Other Strengths And Weaknesses:**

**Strengths:**

This paper is well-written and organized. It presents a novel set of problems to evaluate LLMs, and importantly a mechanism to randomly change 100 questions so that they are always different, thus limiting data contamination during large pre-training runs.

**Weakness:**

Modifying variables in the Variation split does not change the reasoning or final answer of a problem, thus having minimal impact on data contamination.
Changing constants on the other hand can yield to different steps in the reasoning chain and different answers. Nonetheless, the strategy / logic used to solve this problem will be very similar if not the same. As such, the impact on data contamination is limited as LLMs can memorize the “strategy” to solve these exact problems. It is better than LLMs memorizing the raw values and numbers of each variables, but still not as great as novel questions requiring new reasoning strategies. This is of course hard to achieve programmatically, but that is the only way to truly avoid data contamination during finetuning.

Out of the 100 problems selected for Variation only 37 have different constants.

**Questions For Authors:**

What is the distribution of problem difficulty (1-6) in the original & variation splits?

**Relation To Broader Scientific Literature:**

This is a dataset paper. It provides another Math benchmark to evaluate LLMs. This is similar to MATH and GSM8k but the problems are more challenging.
In addition, this paper proposes a subset of questions that can be randomly altered to always provide brand new evaluation questions.

**Theoretical Claims:**

No theoretical claims found.

---

> ### Author Rebuttal · Authors · 2025-04-01
>
> We sincerely thank the reviewer for their thoughtful and detailed response.
>
> > “This is not the correct approach as you already have a metric that will correlate 100% of the time with accuracy, it is accuracy itself. The whole point of using proxy metrics is to evaluate something else than accuracy, i.e. the reasoning chain logic. The correlation between these pseudo metrics should be evaluated against human labels or a strong LLM as a judge such as GPT4o, not against accuracy.”
>
> We thank the reviewer for highlighting the need to evaluate chain-of-thought independently from final-answer accuracy. While we chose a metric that correlates with accuracy (as better reasoning should yield correct answers), we recognize teacher-forced accuracy (TFA) serves primarily as a fine-grained measure of how closely a model follows valid derivations token by token. We acknowledge TFA doesn't necessarily equal reasoning quality without correlation to gold-standard step-by-step evaluations. We will supplement our analysis with human or GPT-4 judgments on intermediate steps to verify whether TFA truly tracks logical consistency rather than just final correctness. These results will be updated during the rebuttal period.
>
> > “The claim that changing variable names, constant values, and turn of phrase will tackle data contamination is not convincing (see explanation in Other Strengths And Weaknesses section).”
>
> We acknowledge that our functional variations don't completely solve data contamination but rather mitigate its effects. Our experimental results show variations consistently decrease accuracy scores across SOTA models, providing strong evidence for mitigation.
> During rebuttal, we fine-tuned DeepSeek-R1 Distill-Qwen-7B (chosen as the best-performing model in its size class) on the full Putnam-AXIOM Original dataset for 5 epochs using full-parameter supervised fine-tuning (SFT) intended to mimic or exaggerate the effects of data contamination. The baseline model achieved 12% accuracy on variations versus 23% accuracy on corresponding originals. After fine-tuning, accuracy increased to 33% on variations versus 80% on corresponding originals. This demonstrates that while contamination allows near-saturation on original problems (+57% accuracy), variations remain challenging (+21% accuracy), confirming their effectiveness against contamination. We plan to extend these experiments to additional models.
>
> > The numbers in Table 1 do not match the scatter plot in Figure 4.
>
> Thank you for pointing out this discrepancy. The accuracy values for TFA in Table 1 and Figure 4 are from a previous dataset version (236 problems), while Table 1 model accuracies are from the expanded dataset (522 problems). For proper comparison, we plotted TFA scores against accuracies using the 236-problem dataset. Due to resource constraints, we haven't yet run TFA on the expanded dataset but are working on this and will provide updated results during the rebuttal phase.
>
> > Why was “perplexity Step” chosen instead of ‘’Common Sense Error’’?  In
> addition, what does it mean to be “chosen” for a pseudo metric if anyway at the
> end of section 4.3 only the top one (TFA) is “selected” to evaluate models (table 1
> & figure 4).
>
> We choose ROSCOE's perplexity step for comparison with our perplexity proxy metric. However we didn't make this selection clear in the paper. The experiments that backed up our rationale for choosing TFA versus baselines is detailed in Section 4.3 of our paper. We did all the experiments to demonstrate our reasoning / rationale to why we chose TFA vs *baselines*. TFA had the highest correlation and performed best on MATH which is why we decided to select it.
>
> > In particular, Table 3 is a pure subset of Table 4 without any other additional
> insight or information. It is therefore unclear what message it provides compared
> to Table 4. Table 4 is probably enough.
>
> We appreciate your suggestion and acknowledge that Table 3 provides no additional insights beyond Table 4. While initially included to highlight the highest-scoring metrics, we are amenable to removing this redundant table in the revised version.
>
> > The caption in Figure 3 may be inverted: right now it shows that the
> accuracies on the Original questions are weaker than on the Variation split.
>
> Thank you for pointing this out, you are correct that the labels for Figure 3 were mistakenly swapped. We will be sure to include the corrected version of the figure in the finalized version.
>
> > What is the distribution of problem difficulty (1-6) in the original &
> variation splits?
>
> Avg difficulty: Original (2.46) vs Variation (2.48) datasets - comparable ratings strengthen our claim that accuracy differences stem from data contamination rather than problem difficulty. Distribution (1-6): Original - 68(1), 109(2), 97(3), 93(4), 75(5), and 80(6) problems; Variation - 8(1), 27(2), 24(3), 11(4), 10(5), and 20(6).

---

### Official Review · Reviewer_Lp2w · 2025-03-13

**Overall Recommendation:** 3

**Summary:**

The paper introduces a new dataset comprised of mathematical problems that have appeared at the Putnam contest for university students. To enable automatic evaluation, the problems are selected or have been reprhased to be such that the solution can be checked automatically as a boxed answer. This rephrasing of the problems need not be semantically preserving, but nonetheless aim to capture the main logical intuition of how the original problem is to be solved. Further, to combat contamination, the authors introduce a functional variation dataset where problems can be generated as (very) similar variants of the original problem by for instance changing constants in the original problem. The paper evaluates a number of open and some proprietary models on these datasets, both in terms of scores, but also via other proxy metrics, in the process demonstrating that current models do not perform well on these tests.

**Claims And Evidence:**

Yes, the paper substantiates the claims made in the introduction/abstract, but some questions are open, which I elaborate on below.

**Essential References Not Discussed:**

Given the mentions of contamination and to better motivate the functional variation dataset, I was hoping to see some experimental evaluation on contamination. of the selected Putnam problems. For instance, using [1].

[1] ConStat: Performance-Based Contamination Detection in Large Language Models, NeurIPS 2024.

**Experimental Designs Or Analyses:**

As mentioned earlier, the selection of the problems and how that is done, is a bit unclear and it is important. Further, I would have hoped that the authors would have done a bit more on the contamination part, that is, checking for contamination of the selected Putnam problems.

**Methods And Evaluation Criteria:**

The evaluation makes sense, however, one concern I have, which the authors can hopefully clarify is: how are the Putnam problems selected exactly? Give the low scores of the models, on problems that are very likely in the training data, this is particularly important. For instance, are some easy problems omitted? Or problems for which it is not trivial to convert it to one with a boxed answer? It is important to understand whether a systematic process is being followed here. Similarly for the functional variation dataset.

**Other Comments Or Suggestions:**

- it was unclear early on whether the rephrasings to get a boxed answer are semantically equivalent; they are not, may be good to state.
- throughout the paper it was unclear how problems are selected, but this was mentioned already above.
- line 313, it is unclear what "to generate evaluation data" means.
- may be I am misreading Fig 3, but I would think original should be scoring higher in accuracy than variation?

**Other Strengths And Weaknesses:**

The datasets introduced by the paper are useful and the paper is well written.

The main concern I have, aside from the ones mentioned already, is the lack of technical depth. For instance, the rephrasings to get a boxed answer are quite straightforward. Similarly for the functional dataset. It would have been interesting to see how other semantically-equivalent text paraphrasings of the problems (many ways to do so), would have affected the results, and similarly for the contamination analysis I mentioned already. Ultimately, the fact that the models are not scoring well on  (slight variants of) problems that are very likely in the dataset, is interesting, and one would have hoped to see more evaluation here.

Still, I believe the datasets can serve as a useful benchmark for future research.

**Questions For Authors:**

Already mentioned in the review.

**Relation To Broader Scientific Literature:**

Obtaining meaningful mathematical benchmarks, ones on which models do not yet work well, is important. The work done to produce this dataset is valuable, and the dataset of Putnam problems, can be useful to develop better models. Similarly, for the functional variation dataset.

**Theoretical Claims:**

This is an experimental paper which introduces a dataset, there are no theoretical claims.

---

> ### Author Rebuttal · Authors · 2025-04-01
>
> We sincerely thank the reviewer for their thoughtful and detailed response. We deeply appreciate the time and effort you invested in providing such valuable feedback.
>
> > ***how are the Putnam problems selected exactly?***
>
> As outlined in Section 3.1, we selected problems based on two primary criteria:
>
> 1. Each problem must yield a definitive final answer that can be enclosed in a \boxed{} format, enabling fully automated evaluation without human grading. These adjustments preserve the original mathematical insights while eliminating the need for human evaluation.
>
> 2. In constructing the dataset, we deliberately excluded binary-answer questions (e.g., true/false or yes/no), as these allow models to succeed via random guessing. These accounted for around 10% of the Putnam problems. This filtering reflects our effort to ensure both rigor and meaningful signal in model performance.
>
> Our datasets excluded questions that were not conducive to these standards, similar to the PutnamBench paper. We curated problems to ensure broad coverage across topics (Algebra, Combinatorics, etc.) and difficulty levels (indicated by 1-6). Based on Putnam’s own difficulty rating we had 205 easy, 182 medium, and 135 hard.
>
> The Variation set is a curated subset of the main dataset, where expert mathematicians systematically alter surface features—like constants or variable names—without changing the problem’s core reasoning. This helps prevent models from relying on memorization.
>
> > ***“experimental evaluation on contamination of the selected Putnam problems.”***
>
> During rebuttal, we fine-tuned DeepSeek-R1 Distill-Qwen-7B (chosen as the best-performing model in its size class) on the full Putnam-AXIOM Original dataset for 5 epochs using full-parameter supervised fine-tuning (SFT) intended to mimic or exaggerate the effects of data contamination. The baseline model achieved 12% accuracy on variations versus 23% accuracy on corresponding originals. After fine-tuning, accuracy increased to 33% on variations versus 80% on corresponding originals. This demonstrates that while contamination allows near-saturation on original problems (+57% accuracy), variations remain challenging (+21% accuracy), confirming their effectiveness against contamination. We plan to extend these experiments to additional models.
>
> > ***lack of technical depth...***
>
> Our goal was not to complicate the problem statements themselves but rather to preserve the intrinsic difficulty of the Putnam examination while making the benchmark automatically gradable at scale. The “boxed-answer” approach, though seemingly straightforward, ensures that the dataset is both substantial and tractable for automated evaluation—an approach shared by similar math benchmarks.
>
> > ***“how other semantically-equivalent text paraphrasings...”***
>
> Thank you for pointing out this area of exploration. We tried paraphrasing the variation questions with the assistance of LLMs and meticulously verified by humans. This resulted in minimal changes in accuracy and thus didn’t report paraphrasing for simplicity. We will clarify this point in our revised submission.
>
> > ***the fact that the models are not scoring well on...***
>
> We compared model responses between the Original and Variation datasets. Both responses adopted a “step by step” narrative style. However, original responses tend to follow a systematic, clear approach while responses often reveal internal doubts result in incomplete algebraic details or solutions. One such example is 1947 B4 where the question is:
>
>  > Given $P(z) = z^2 + az + b$, a quadratic polynomial of the complex variable $z$...Find the values of $a$ and $b$ and give their sum.
>
> Original Solution (shortened for readability):
> > Let's think step by step. Since $|P(z)| = 1$ for all $|z| = 1$, then for $z$ on the unit circle, $|z^2 + az + b| = 1$. If $z \\neq 0$, then $|1 + a/z + b/z^2| = 1$, so $|Q(w)| = 1$ where $w = 1/z$ and $Q(w) = w^2 + a w + b$...Therefore, the sum $a + b = b - 1/b$. But since $|b|=1$, we have $b = e^{i\\phi}$ for some angle $\\phi$, so $a + b = e^{i\\phi} - e^{-i\\phi} = 2i \\sin \\phi$.
>
> Variation Solution (shortened):
> > Let's think step by step. Since $P(p)$ is a quadratic polynomial and $|P(p)| = 88$ for all $|p|=88$...Okay, so I've got this problem here where I need to find the angle A in triangle ABC. The conditions given are a bit specific: angle A is less than angle C...So,180 - \u03b1/2 > 90 Which gives, \u03b1/2
>
> The original solution directly addresses the problem, while the variation’s solution “hallucinates” a triangle ABC scenario and omits key steps. We see three main causes:
> 1. LLMs often appear to recall answer patterns rather than genuinely solve problems
> 2. Small changes to variable names or constants (e.g., from *x* to *p*) can disrupt these memorized patterns, suggesting a lack of true variable abstraction.
> 3. Unusual symbol choices or number pairings can misalign the prompt, triggering irrelevant or hallucinatory reasoning steps.

---

### Official Review · Reviewer_6ec1 · 2025-03-24

**Overall Recommendation:** 4

**Summary:**

Putnam-AXIOM is a new benchmark designed to assess higher-level mathematical reasoning in large language models (LLMs), using 522 challenging problems from the William Lowell Putnam Mathematical Competition. To address data contamination, the authors introduce functional variations of 100 problems by altering variables, constants, and phrasing, enabling infinite novel yet equally difficult problems. Evaluation shows that even top models like GPT-4o and o1-preview perform significantly worse on these variations, highlighting the limitations of current models. The benchmark also proposes new reasoning metrics like Teacher-Forced Accuracy (TFA) to go beyond final boxed answers and better capture reasoning depth.

**Claims And Evidence:**

Yes. This paper mainly has two claims: 1. variations of the Putnam benchmark are likely to cause trouble to existing model (due to issues like data contamination); 2. the proposed teach-forced accuracy (TFA) is better correlated with boxed accuracy than existing metrics like ROSCOE. To me, both claims have been empirically validated.

**Essential References Not Discussed:**

N/A

**Experimental Designs Or Analyses:**

The experimental designs look good to me. I especially appreciate the comprehensive ablation study in the appendix.

**Methods And Evaluation Criteria:**

Yes. Putnam is a solid source of dataset for measuring mathematical reasoning. The effectiveness of the newly proposed TFA metric has also been validated on the widely used MATH dataset.

**Other Comments Or Suggestions:**

- Figure 3 may have the labels for Original and Variation swapped. Based on my understanding and the data in Table 2, the blue intervals should correspond to the Variation set.

**Other Strengths And Weaknesses:**

- I’d love to see some discussion on to what extent can TFA—or any of the other proposed metrics—address the issue of correct answers with flawed reasoning, which is notably common in leading models like o1.

**Questions For Authors:**

N/A

**Relation To Broader Scientific Literature:**

Both the new dataset and the automatic proxy metric offer valuable contributions to mathematical reasoning research, especially as many existing benchmarks have become saturated.

**Theoretical Claims:**

N/A

---

> ### Author Rebuttal · Authors · 2025-04-01
>
> We thank the reviewer for their thoughtful and detailed review, particularly for recognizing that our work addresses an important gap by introducing a benchmark capable of measuring higher-level mathematical reasoning, and appreciating the thoroughness of our experimental designs and comprehensive ablation studies. Your positive feedback about the validity and utility of our claims and metrics is greatly valued.
>
> > ***Figure 3 may have the labels for Original and Variation swapped***
>
> Thank you for pointing this out, you are correct that the labels for Figure 3 were mistakenly swapped. We will be sure to include the corrected version of the figure in our revised submission.
>
> > ***discussion on to what extent can TFA—or any of the other proposed metrics—address the issue of correct answers with flawed reasoning, which is notably common in leading models like o1***
>
> We appreciate the reviewer’s interest in how Teacher-Forced Accuracy (TFA) handles scenarios in which the model’s final answer is correct despite a flawed chain of thought. TFA differs from standard auto-regressive (AR) generation because each predicted token is conditioned on the gold-standard solution, thus focusing on whether the model *recognizes* a correct reasoning path rather than whether it can generate one autonomously. From a different perspective, it uses the forward during training instead of during inference for generation and then computes per token accuracy match to the gold reference string.
>
> Because TFA never lets the model produce its own possibly incorrect steps, it does not directly reveal flawed intermediate reasoning in AR generation. This is one reason we were motivated to rigorously verify TFA’s validity – by analyzing its correlation with boxed-answer accuracy — ensuring that it reliably measures deeper competence across multiple problems (MATH, Putnam-AXIOM), fourteen models and six model families.
>
> Indeed, one could explore hybrid approaches —for instance, partially teacher-forcing the solution, then auto-regressively generating the remainder—but these methods are often computationally expensive or require custom hardware code. In line with Occam’s Razor, we opted for TFA’s simpler, scalable approach first.
>
> We will clarify these points in the revised manuscript, adding discussion in the TFA section and appendix to make explicit the difference between TFA and AR generation.
>
> # Evaluating the Effectiveness of Variations on Contamination
> During the rebuttal phase, we have performed our own fine-tuning experiment with DeepSeek-R1 Distill-Qwen-7B (we chose this model because it was the best-performing model of its parameter size in Table 2). On the Putnam-AXIOM Variation dataset, this baseline model received 12% accuracy on the variation questions and 23% accuracy on the corresponding original questions.
>
> We performed full-parameter SFT (intended to mimic or exaggerate the effects of data contamination) on the full Putnam-AXIOM Original dataset (with all 522 questions), running for 5 epochs. We then evaluated the fine-tuned model on Putnam-AXIOM Variations (with 100 variation questions and 100 corresponding original questions). After fine-tuning, the model received 33% accuracy on the variations and 80% accuracy on the corresponding original questions.
>
> Clearly, the full-parameter SFT successfully contaminated the model such that it was able to attain saturation on the corresponding original problems with a 57% increase in accuracy; however, the variations still proved to be challenging for the model to solve, given a mere 21% increase in accuracy. Our experiment shows that our variations are a useful tool in combating data contamination.

---

### Decision · Program_Chairs · 2025-05-01

**Decision:**

Accept (poster)

**Comment:**

This paper received unanimous accepts from reviewers. The authors are encouraged to incorporate the reviewers' suggestions in the camera-ready version.